# Prognostic Role of Preoperative Neutrophil-To-Lymphocyte Ratio (NLR) and Recurrence at First Evaluation after Bacillus Calmette–Guérin (BCG) Induction in Non-Muscle-Invasive Bladder Cancer

**DOI:** 10.3390/diagnostics13193114

**Published:** 2023-10-02

**Authors:** Junghoon Lee, Sangjun Yoo, Min Soo Choo, Min Chul Cho, Hwancheol Son, Hyeon Jeong

**Affiliations:** 1Department of Urology, Seoul National University Seoul Metropolitan Government Boramae Medical Center, Seoul National University College of Medicine, Seoul 07061, Republic of Korea; deftblow@gmail.com (J.L.);; 2Institute for Peace and Unification Studies, Seoul National University, Seoul 08826, Republic of Korea

**Keywords:** non-muscle-invasive bladder neoplasms, BCG induction, non-complete response, BCG failure, neutrophil-to-lymphocytes ratio

## Abstract

We investigated the prognosis of BCG induction-only treatment and non-complete response (CR) at the first 3-month evaluation and examined factors associated with CR. In total, 209 patients with moderate- and high-risk NMIBC who received BCG induction-only treatment between 2008 and 2020 were retrospectively analyzed. Recurrence-free survival (RFS) and progression-free survival (PFS) were assessed based on the initial NMIBC stage. PFS and associated factors of non-CR compared to CR were also assessed. Initial T1 high-grade (HG) (*n* = 93) had poorer RFS and PFS after BCG induction-only treatment than Ta low-grade (LG) (*p* = 0.029, *p* = 0.002). Non-CR (*n* = 37) had a different neutrophil-to-lymphocyte ratio (NLR) (2.81 ± 1.02 vs. 1.97 ± 0.92) and T staging from CR (*p* < 0.001, *p* = 0.008). T1HG recurrence was associated with a worse PFS compared to non-T1HG (13.7 months vs. 101.7 months, *p* < 0.001). There was no difference in PFS between T1HG and T1LG. T1 and NLR were predictors of response at 3 months in multivariable analysis (*p* = 0.004, *p* = 0.029). NLR was also found to be an associated factor with RFS and PFS of bladder cancer (*p* < 0.001, *p* < 0.001). BCG induction-only treatment was effective for high-risk TaLG but not for T1HG. T1HG recurrence at 3 months after BCG induction has a poor prognosis for bladder cancer. Preoperative NLR and T1 were predictors of non-CR, and NLR was also associated with the long-term prognosis of bladder cancer.

## 1. Introduction

Bacillus Calmette–Guérin (BCG) treatment is known to be effective in reducing the recurrence and progression of moderate-to-high-risk non-muscle-invasive bladder cancer (NMIBC) [1,2]. BCG therapy consists of induction treatment after transurethral resection of bladder tumor (TURBT) and maintenance treatment over the long term [1,3]. Some studies reported that additional maintenance treatment is more effective in reducing the recurrence and progression of bladder cancer compared to induction-only treatment. However, another study has also reported that there is no significant difference in the effectiveness of these two approaches [4]. In addition, long-term BCG maintenance treatment has some limitations that make it difficult to provide therapy adequately [5,6,7]. The limitations of BCG treatment include poor patient compliance, the occurrence of side effects, and the ongoing global BCG shortage. As a result, in clinical practice, patients sometimes face situations where they receive only limited treatments, such as induction-only therapy. Therefore, further research is needed to identify the appropriate patient population for induction treatment and its effectiveness in situations where BCG maintenance is limited.

BCG failure, such as BCG-refractory, BCG-unresponsive, and BCG relapse, is not well defined and varies across studies in terms of classification, timing of assessment, and pathologic conditions [1,2,8]. To address this issue, the FDA recently defined “BCG-unresponsive” to provide guidance for the development of drugs and clinical trials [9]. The guidelines include the presence of T1 high-grade (HG) at the first evaluation after induction BCG as one of the BCG-unresponsive criteria. In some studies, patients with non-complete response (CR) observed any tumor recurrence on the first cystoscopy evaluation at 3 months after BCG induction had a poor prognosis [10,11,12]. Further studies are currently recommended to define the status and describe the natural history of bladder cancer according to response at 3 months after BCG induction [8].

Since the main mechanism of action of BCG is thought to be via immune responses, studies using factors reflecting the patient’s immune status to predict the effect of BCG treatment have been recently reported [13,14,15]. Recently, studies have reported that the prognosis of various diseases is related to serum inflammatory markers such as the neutrophil-to-lymphocyte ratio (NLR), platelet-to-lymphocyte ratio (PLR), and lymphocyte-to-monocyte ratio (LMR) [16,17,18,19]. There is some evidence suggesting that these inflammatory markers reflect the recurrence and progression of bladder cancer [20,21,22,23]. However, the association between the serum inflammatory marker and the short-term response of BCG induction has not been studied yet.

Considering the limitations in BCG treatment such as poor compliance, side effects, and worldwide shortages, identification of early response and prognosis after BCG induction-only treatment can clinically improve the efficiency and effectiveness of bladder cancer treatment. The association between serum inflammatory markers (such as NLR, PLR, and LMR) and the effect of BCG induction could provide evidence for the effect of patient immune status on short-term BCG immunotherapy. This study aimed to investigate the efficacy of BCG induction-only treatment and the prognosis of non-CR at the first evaluation after BCG induction. Additionally, we aimed to identify predictors, including NLR, PLR, and LMR, for non-CR and to determine the association of serum inflammatory biomarkers with the prognosis of bladder cancer after BCG treatment.

## 2. Materials and Methods

### 2.1. Study Sample

The study was conducted in accordance with the Declaration of Helsinki and approved by the Institutional Review Board of Seoul National University-Seoul Metropolitan Government Boramae Medical Center (No. 20-2021-95). We retrospectively analyzed a total of 273 patients who underwent BCG induction for intermediate- or high-risk NMIBC after TURBT from January 2008 to April 2020. High-risk NMIBC was defined as HG, presence of carcinoma in situ (CIS), T1 or recurrent, multiple, and ≥3 cm of Ta low-grade (LG) [1,2]. Intermediate-risk NMIBC was defined as recurrent, multiple, or ≥3 cm of Ta LG. Exclusion criteria were failure to complete 6 BCG induction cycles, incomplete resection of tumor during initial TURBT, detected persistent urothelial cancer in the second TURBT, loss to follow-up before first cystoscopy after BCG induction, and presence of immune disease or concurrent other malignancy.

### 2.2. Study Design

The BCG induction protocol was started within 2 to 6 weeks after initial TURBT and involved 6 cycles of weekly intravesical instillation of the TICE BCG strain (Onco-Tice 12.5 mg). Recurrence was defined as any tumor diagnosed pathologically during follow-up. Progression was defined as the development of muscle-invasive or metastatic bladder cancer [8]. Short-term response at the first cystoscopy performed 3 months after BCG induction was categorized as CR and non-CR. We defined CR as patients with no evidence of tumor and non-CR as patients with gross and pathological confirmation of any bladder cancer [8,10,11,12]. Non-CR patients with T1 HG at 3 months were subclassified as BCG-refractory [1,8]. Age, sex, primary tumor stage, grade, concurrent carcinoma in situ (CIS), size and multiplicity of tumors, and preoperative NLR within 3 months of first TURBT as serum inflammatory marker were identified as the clinicopathological characteristics of the patients. NLR values closest to the date of surgery were selected.

### 2.3. Outcome Measurements

The primary endpoint was to assess the response to BCG induction-only treatment by pathologic stage and the prognosis of CR. We conducted a long-term evaluation of the response to BCG induction by comparing recurrence-free survival (RFS) and progression-free survival (PFS) based on pathological staging after primary TURBT in patients with BCG induction treatment only. We compared the PFS of CR and non-CR after BCG induction. PFS was also compared between BCG-refractory and non-BCG-refractory patients in non-CR.

The secondary endpoint was to determine whether perioperative factors, including the patient’s immune status, affect the short-term BCG response and prognosis of bladder cancer. Perioperative factors that could predict non-CR were analyzed, and RFS and PFS according to NLR in BCG induction treatment patients were evaluated.

### 2.4. Statistical Analyses

Continuous variables were presented as the mean and standard deviation, and non-continuous variables were presented as frequency count and percentage. For comparison of continuous and non-continuous variables, the Mann–Whitney U-test and Pearson’s chi-square tests were performed, respectively. The Kaplan–Meier curves were used to calculate RFS and PFS. Multivariable analysis used logistic regression to investigate the predictors of non-CR after BCG induction using the forward stepwise selection method for factors with *p* < 0.10 among univariable analysis covariates. The receiver operating characteristic (ROC) curve with the Youden index was used to confirm the diagnostic accuracy and cutoff value of NLR. Statistical significance was set at *p* < 0.05. All statistical analyses were conducted using the SPSS program, version 22.0 (IBM, Armonk, NY, USA).

## 3. Results

### 3.1. Clinicopathologic Characteristics of the Patients

A total of 209 patients (174 males and 35 females) were included in the analysis. The mean age of the patients was 72.10 ± 8.41 years. Patients had a mean follow-up time of 57.71 ± 33.90 months. Pathology stages were categorized into Ta LG 47 (22.5%), Ta HG 33 (15.8%), T1 LG 13 (6.2%), T1 HG 93 (44.5%), and Tis 23 (11.0%).

Within the 5–7-month follow-up period after treatment, 24 patients (11.5%; primary tumor: 6 Ta LG, 5 Ta HG, 0 T1 LG, 10 T1 HG, 3 Tis) experienced recurrence. Additionally, 7 patients (3.3%; primary tumor: 2 Ta HG, 5 T1 HG) showed disease progression. Subsequently, within 8–12 months, 15 patients (7.2%; primary tumor: 3 Ta LG, 1 Ta HG, 4 T1 LG, 7 T1 HG, 1 Tis) experienced recurrence, and 11 patients (5.3%; primary tumor: 1 Ta LG, 2 Ta HG, 0 T1 LG, 7 T1 HG, 1 Tis) showed disease progression.

After BCG induction, 37 (17.7%) non-CR patients (BCG-refractory: *n* = 28, non-BCG-refractory: *n* = 9) had bladder cancer at 3 months after BCG induction. The initial T stage of these non-CR patients was T1 28 (75.7%), Ta 3 (8.1%), and Tis 6 (16.2%), which was significantly different from that of T1 78 (45.3%), Ta 77 (44.8%) and Tis 17 (9.9%) with CR (*p* = 0.008). Preoperative NLR showed differences among those with non-CR and CR (2.81 ± 1.02- vs. 1.97 ± 0.92, *p* < 0.001). Table 1 summarizes the clinicopathological characteristics of the patients.

### 3.2. Analysis of Recurrence and Progression in Patients Treated with BCG Induction-Only by Pathologic Stage after Primary TURBT

The mean time to recurrence for each stage was as follows: Ta LG (15.37 ± 13.89 months), Ta HG (18.54 ± 13.56 months), T1 LG (14.64 ± 17.02 months), T1 HG (11.97 ± 12.83 months), and Tis (14.32 ± 22.15 months). The mean time to progression was as follows: Ta LG (31.78 ± 42.79 months), Ta HG (28.74 ± 26.27 months), T1 LG (23.98 ± 12.75 months), T1 HG (23.00 ± 22.29 months), and Tis (61.22 ± 68.19 months).

T1 HG tumors do not respond adequately to BCG induction-only, and high-risk Ta LG presented favorable responses in terms of recurrence and progression. The average RFS differed across tumor stages, with Ta LG at 51 months, Ta HG at 67 months, T1 LG at 58 months, T1 HG at 50 months, and Tis at 41 months. T1 HG showed significant differences compared to Ta LG and Ta HG (*p* = 0.029, *p* = 0.009, respectively) (Figure 1A). The average PFS varied among different stages, with Ta LG at 126 months, Ta HG at 82 months, T1 LG at 111 months, T1 HG at 102 months, and Tis at 118 months (Figure 1B). Significant differences in PFS were observed between T1 HG and Ta LG (*p* = 0.002) and between Ta HG and Ta LG (*p* = 0.013). However, there was no significant difference in PFS between T1 HG and Ta HG (*p* = 0.580).

### 3.3. Analysis of Progression-Free Survival in Patients with Non-Complete Response

A total of 123 patients recurred with a mean time of 13.8 ± 14.7 months, and 70 patients experienced progression with a mean time of 19.2 ± 18.2 months. Non-CR had a significantly poorer prognosis for PFS than CR (*p* < 0.001) (Figure 2A). The mean PFS was 38.0 months of non-CR, while 101.8 months of CR. In the subgroup analysis of non-CR, there was a significant difference in PFS of 13.7 months in BCG-refractory and 101.7 months in non-BCG-refractory (*p* < 0.001) (Figure 2B). There was no significant difference in PFS between T1 HG and T1 LG, and a significant difference between Ta HG and Ta LG (*p* = 0.125, *p* = 0.002, respectively). Therefore, recurrence of T1 or HG at 3 months after BCG induction was associated with a poor prognosis.

### 3.4. Predictors of Non-Complete Response

In univariable logistic regression analysis, the factors related to non-CR were analyzed to be significant for T1 stage, grade, NLR, PLR, and LMR. Age > 70, male sex, positive malignancy of urine cytology, grade of tumor, and presence of CIS were not significantly associated with non-CR (Table 2). When multivariable analysis was performed on the factors with *p* < 0.1 on univariable logistic regression analysis, T1 and NLR were significant factors that could predict non-CR after BCG induction (*p* = 0.004, *p* = 0.029, respectively).

### 3.5. Association of Preoperative NLR with BCG Response or Prognosis of Bladder Cancer

The ROC curve analysis for NLR showed the highest predictive power with 73% sensitivity and 77% specificity for non-CR with an optimal cutoff value of NLR at 2.42 (area under the curve (AUC) = 0.761, 95% confidence interval (CI)= 0.667–0.845, *p* < 0.001) (Figure 3). The RFS after BCG induction was significantly different, with a mean of 33.7 months for patients with NLR > 2.42 and a mean of 66.0 months for patients with NLR ≤ 2.42 (*p* < 0.001) (Figure 4A). And patients with NLR > 2.42 had a significantly poorer prognosis in PFS than patients with NLR ≤ 2.42 (mean 45.8 months vs. 108.0 months, *p* < 0.001) (Figure 4B).

## 4. Discussion

The evaluation and prediction of response after BCG induction will help to select appropriate patients and to determine whether to perform salvage BCG treatment or alternative treatments such as early cystectomy or new drugs like Pembrolizumab. Furthermore, efforts to clearly define BCG response for establishing appropriate design settings in clinical trials are also required in preparation for the era of new drugs. The limitations of BCG treatment must also be taken into consideration in the course of treatment [5,6,7]. One of the major challenges associated with BCG treatment is the global shortage of BCG, which is causing significant delays and disruptions in patients receiving treatment. In addition, patients undergoing BCG maintenance are required to visit the clinic for prolonged periods, which can be physically and emotionally challenging. The discomfort associated with procedures such as catheterization also contributes to a gradual decline in patient compliance. Furthermore, some side effects that may occur during BCG maintenance, such as sepsis and irritative voiding symptoms, are also a concern with the long-term use of BCG.

In this study, the response to BCG induction-only treatment in terms of bladder cancer recurrence and progression was particularly insufficient in T1 HG compared to other staging groups. Moreover, the PFS of non-CR at first evaluation after BCG induction was poor. Among non-CRs, T1 HG was particularly associated with a worse prognosis than LG, Tis, or Ta recurrence. T1 stage and NLR were predictors of response for the 3-month outcome after BCG induction. And NLR may also be a factor that reflects the long-term prognosis of bladder cancer.

Whether BCG induction-only treatment or additional treatment for more than 1 year is more beneficial is still a controversial topic [4]. Many studies did not find a significant difference between the two treatment methods in lowering the recurrence of bladder cancer. A review of the literature published in 2018 analyzed 10 randomized controlled trials involving 1402 patients, but there was no significant difference in the recurrence rate between the induction-only treatment and the additional maintenance treatment (45.78% vs. 36.13%, *p* = 0.026) [4]. The CUETO study compared 191 induction-only arms and 200 3-year maintenance arms through randomized controlled trials but showed no difference between disease relapse at 5 years (33.5% vs. 38.5%, respectively) and disease-free interval (hazard ratio (HR)0.83; 95% CI 0.61–1.13; *p* = 0.2). There was also no difference between the two groups in progression rate at 5 years (16% vs. 19.5%, respectively) and time to progression (HR 0.79; 95% CI 0.50–1.26; *p* = 0.3). However, Lamm et al. reported a CR of 69% at 6 months when only BCG induction was performed, whereas the additional 3-week maintenance group reported a significant difference of 84% (*p* < 0.01) [24].

There are only a few studies on the rapid recurrence of non-CR after BCG induction. Mmeje et al. reported that 84 (9.2%) of 917 BCG induction patients had papillary recurrence excluding CIS at the 3-month evaluation [12]. Another study reported that 41 patients (21.5%) had non-CR at 3 months after intravesical therapy, and non-CR was also a predictive factor for progression [10]. Regarding prognosis, Southwest Oncology Group (SWOG) analyzed 434 patients with BCG induction and reported that the 5-year survival probabilities of patients who achieved a CR and those who did not achieve a CR at the first endoscopy were 77% and 62%, respectively [11]. In addition, patients without CR had a 67% higher mortality rate (HR of CR = 0.60; 95% CI 0.44–0.81; *p* < 0.001). This study found that 31 non-CR patients (17.9%) were detected at the first cystoscopy evaluation and showed worse PFS compared to CR (38.0 months vs. 101.8 months, *p* < 0.001). And among non-CR patients, PFS was poorer for T1 HG recurrence than for Tis, Ta, or LG recurrence (13.7 months vs. 101.7 months, *p* < 0.001). There are not enough studies on the prognosis of Ta, LG, and Tis recurrence after BCG induction, and heterogeneous results have been reported. One study reported that Ta HG recurrence at 3 months after BCG induction had a similar 5-year PFS to that of T1 recurrence (77% vs. 83%) [12]. On the other hand, Ta LG recurrence (*n* = 13) had no bladder cancer progression. Steinberg et al. reported no significant difference in the association of Ta vs. T1 recurrence with treatment failure in patients with BCG failure using univariate analysis (*p* = 0.45) [25]. Planning further treatment by focusing on patients with T1 HG recurrence at the first evaluation would improve the efficiency of BCG treatment.

It has been reported that patient immune status is associated with the prognosis of bladder cancer [26,27]. In our data, the preoperative NLR showed significant differences between the CR and non-CR groups in BCG induction. Additional multivariable analysis showed that NLR was significant as a predictor of non-CR, and the cutoff value was 2.42 for the highest predictive power. A significant difference in the prognosis of NMIBC was observed according to an NLR of 2.42. Vartolomei et al. reported in a meta-analysis that a high preoperative NLR increases the risk of recurrence and progression of NMIBC and is a predictor of recurrence and progression after BCG treatment [20]. They described that 6 studies involving 2298 participants with NMIBC reported that an increased NLR was associated with decreased RFS (pooled HR = 1.78; 95% CI: 1.32–2.4, *p* = 0.001) and PFS (pooled HR = 2.14; 95% CI: 1.59–2.87, *p* = 0.001). Another study conducted as a cohort study of 113 patients analyzed the association between NLR and bladder cancer recurrence [23]. A total of 64 patients (56.6%) showed recurrence in the median time of 9 months, and multivariable analysis showed that NLR > 2.5 increased the risk of recurrence and worsened RFS in patients treated with BCG. To date, NLR is the most commonly used representative serum inflammatory marker in the analysis of the association between patient immune status and diseases. Studies on PLR and LMR as serum inflammatory markers of bladder cancer have also been reported. A study published in 2021 analyzed NLR, PLR, and LMR as independent prognostic markers of progression in 125 BCG-treated patients (*p* < 0.05) [28]. In the ROC curve analysis, the addition of the LMR to the baseline model showed an increase of 0.08 in AUC (*p* = 0.001), but the addition of the LNR and PLR factors did not show a significant increase in AUC compared to the baseline model. However, there are still insufficient studies on PLR and LMR in bladder cancer.

By understanding the immunologic characteristics of NLR and bladder cancer, it is possible to interpret how NLR functions as a predictor of prognosis in bladder cancer. In the tumor microenvironment, neutrophils play a role in promoting tumor progression by suppressing the anti-cancer immune system such as cytotoxic T lymphocytes and NK (natural killer) cells, as well as contributing to processes like angiogenesis and tumor cell proliferation [29]. Conversely, lymphocytes infiltrate tumors and have an anti-tumor role [30]. Therefore, NLR has been considered a biomarker that reflects systemic inflammatory responses (SIR) by utilizing these mechanisms to evaluate tumor activity. The growth of tumors is driven by a microenvironment that creates conditions favorable for tumorigenesis [31]. This microenvironment involves various factors, including intrinsic inflammatory responses, angiogenesis, immune suppression, and cellular changes. Increased activity of cancer cells leads to SIR, and SIR in turn further promotes cancer growth through activation of the pro-tumor responses, creating a vicious cycle. Bladder cancer has been reported to be highly immunogenic, evading normal immune responses [32]. Considering the relationship between bladder cancer and immune response, NLR may be associated with bladder cancer long-term prognosis even with BCG maintenance therapy [33]. NLR is known to have the advantages of being cost-effective, easy to perform, and less sensitive to physiological changes such as dehydration or physical activity compared to other indexes such as white blood cell count [34]. Therefore, NLR is expected to be a valuable factor for predicting the prognosis of bladder cancer.

Our study has several limitations. First, this was a retrospective study conducted with a small sample size. However, there have been no studies on the relationship between NLR and response of BCG induction. There is also limited research on the prognosis of bladder cancer after BCG induction-only treatment. Therefore, it is expected that the clinical interest induced by our study will lead to large-scale, well-designed studies. Second, the definition and classification of BCG failure have not yet been fully established [1,2], each previous study on BCG failure also had differences in the study population. In some studies, CR and non-CR are used as one of the criteria to classify BCG failure. Third, due to the limitations of long-term BCG treatment, a small sample size, and the retrospective nature of the study, this research did not have a comparison group for BCG induction-only therapy. Nevertheless, an understanding of the response to BCG induction based on the stages of primary TURBT allowed us to speculate on the appropriate patient group. A large-scale, well-designed BCG treatment study and a clear definition of BCG failure are needed to improve the prognosis of bladder cancer.

## 5. Conclusions

BCG induction-only treatment could be cautiously considered for high-risk Ta LG patients when long-term BCG treatment options are limited. Short-term response after BCG induction and patient immune status are key factors in predicting bladder cancer prognosis. Non-CR at 3 months after BCG induction had a worse PFS compared to CR in NMIBC. In particular, T1 or HG recurrence is responsible for the poor prognosis and represents a disease status that should be considered for further treatment. Preoperative NLR and T1 were significant factors predicting non-CR after BCG induction treatment in patients with NMIBC. Additionally, NLR was a serum inflammatory marker that could identify patients with worse prognosis in RFS and PFS after BCG induction.

## Figures and Tables

**Figure 1 diagnostics-13-03114-f001:**
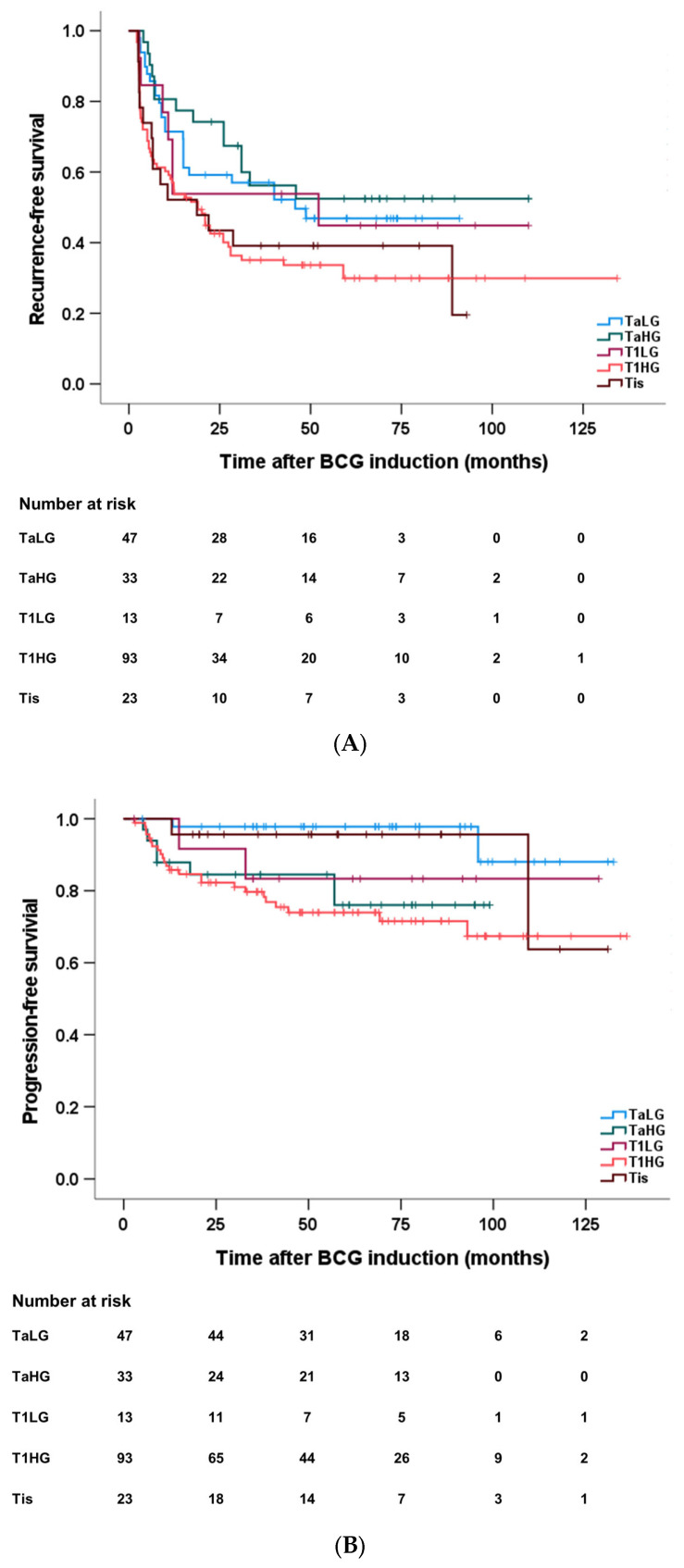
The difference in (**A**) RFS based on initial stages was compared (Ta LG 51 months, Ta HG 67 months, T1 LG 58 months, T1 HG 50 months, and Tis 41 months). T1 HG demonstrated a significant difference in RFS compared to Ta LG and Ta HG (*p* = 0.029, *p* = 0.009, respectively). (**B**) PFS based on initial stages was also compared (Ta LG at 126 months, Ta HG at 82 months, T1 LG at 111 months, T1 HG at 102 months, and Tis at 118 months). There was a significant PFS difference between T1 HG and Ta LG (*p* = 0.002) and Ta HG and Ta LG (*p* = 0.013). T1 HG and Ta HG did not differ in PFS (*p* = 0.580). LG = low-grade, HG = high-grade, RFS = recurrence-free survival, PFS = progression-free survival.

**Figure 2 diagnostics-13-03114-f002:**
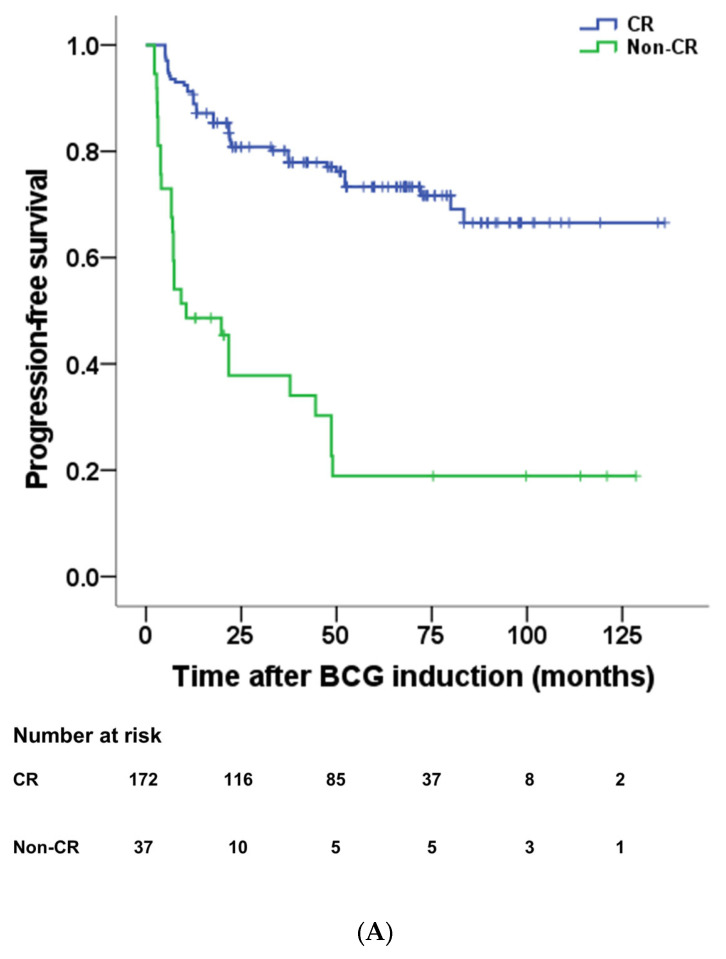
The difference in progression-free survival (**A**) between non-CR and CR at 3 months after BCG induction was compared (38.0 months vs. 101.8 months, *p* < 0.001). Comparison of progression-free survival (**B**) between T1 HG recurrence and Tis, Ta, or LG recurrence in non-CR patients was also analyzed (13.7 months vs. 101.7 months, *p* < 0.001). CR = complete response, BCG = Bacillus Calmette–Guérin.

**Figure 3 diagnostics-13-03114-f003:**
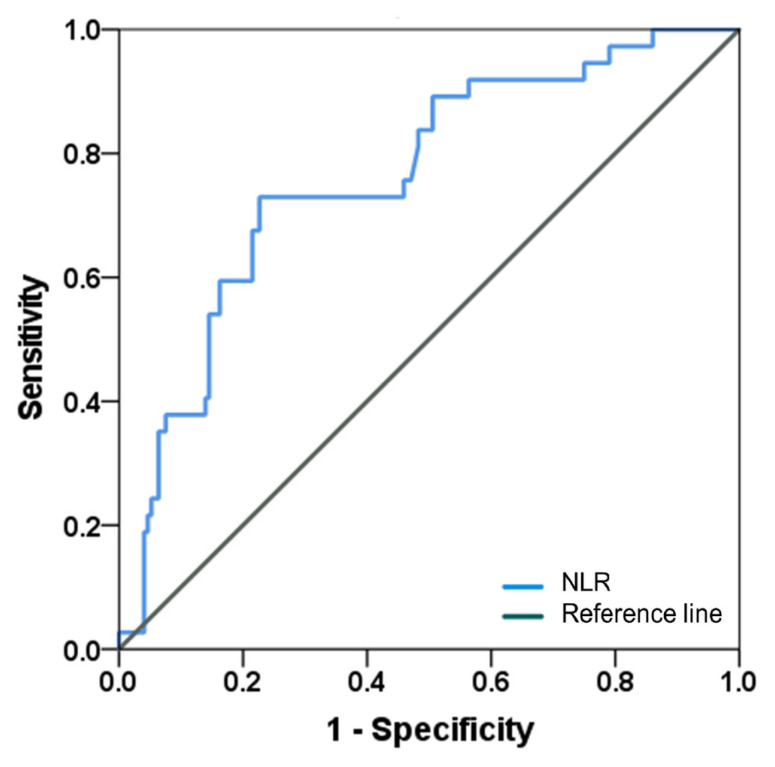
Receiving operator characteristics curve of NLR (AUC = 0.761, 95% CI: 0.667–0.845) and cutoff value of NLR at 2.42 (sensitivity = 73%, specificity = 77%). NLR = neutrophil-to-lymphocyte ratio, AUC = area under the receiver operating characteristic curve.

**Figure 4 diagnostics-13-03114-f004:**
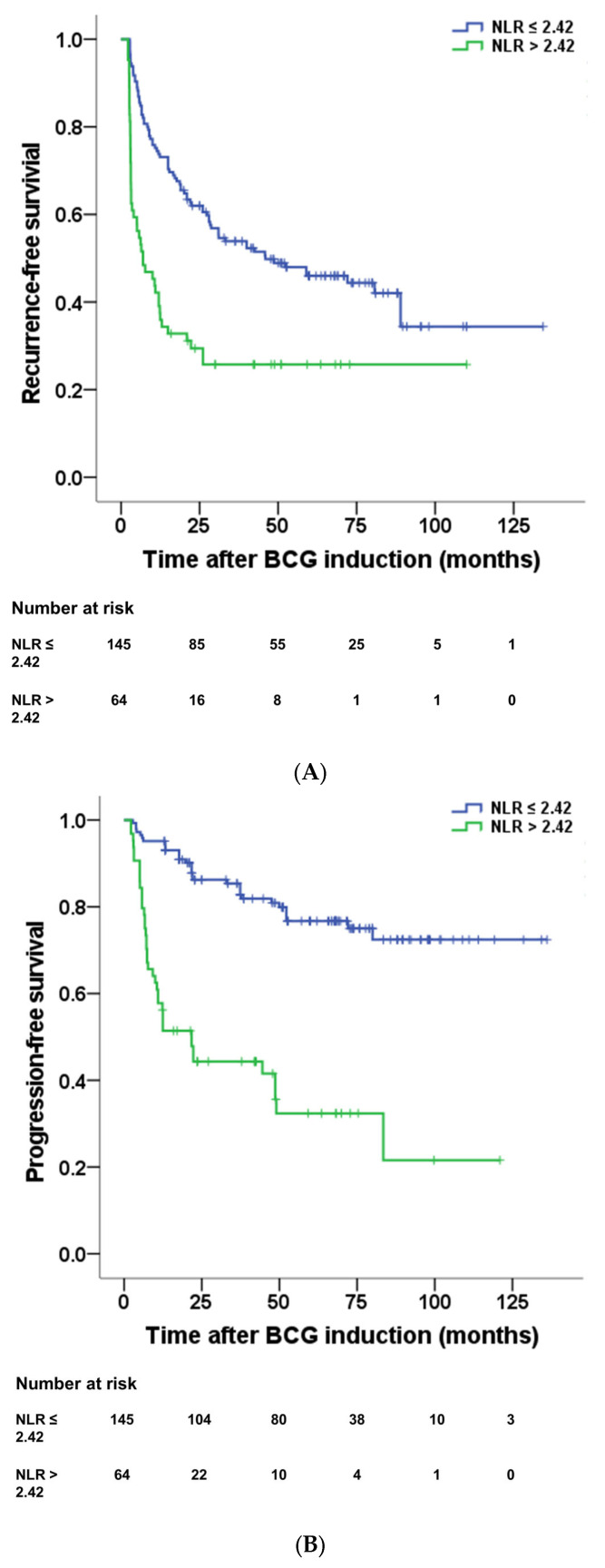
The difference between (**A**) recurrence-free survival (mean 33.7 months vs. 66.0 months, *p* < 0.001) and (**B**) progression-free survival (45.8 months vs. 108.0 months, *p* < 0.001) after BCG induction was analyzed according to NLR 2.42. Patients who received BCG induction treatment with NLR > 2.42 had a poor prognosis. BCG = Bacillus Calmette–Guérin, NLR = neutrophil-to-lymphocyte ratio.

**Table 1 diagnostics-13-03114-t001:** Patient characteristics of recurrence at first follow-up cystoscopy after BCG induction.

	Total	Complete Response	Non-Complete Response	*p*
Number of patients, *n* (%)	209 (100.0)	172 (82.3)	37 (17.7)	<0.001
Age at BCG, years, mean ± SD	72.10 ± 8.41	71.99 ± 8.47	72.61 ± 8.18	0.600
Male, *n* (%)	174 (83.3)	142 (82.6)	32 (86.5)	0.562
Diabetes, *n* (%)	54 (25.8)	46 (26.7)	8 (21.6)	0.518
BMI, kg/m^2^, mean ± SD	24.46 ± 3.25	24.40 ± 3.17	24.74 ± 3.61	0.566
Prev TURBT, mean ± SD	1.31 ± 1.93	1.32 ± 1.78	1.24 ± 2.53	0.258
T stage, *n* (%)				0.008
Ta	80 (38.3)	77 (44.8)	3 (8.1)	
T1	106 (50.7)	78 (45.3)	28 (75.7)	
Tis	23 (11.0)	17 (9.9)	6 (16.2)	
Grade, *n* (%)				0.018
Low	62 (29.7)	57 (33.1)	5 (13.5)	
High	147 (70.3)	115 (66.9)	32 (86.5)	
Presence of CIS, *n* (%)	33 (15.8)	26 (15.1)	7 (18.9)	0.565
Size, *n* (%)				0.328
≤3 cm	180 (86.1)	150 (87.2)	30 (81.1)	
>3 cm	29 (13.9)	22 (12.8)	7 (18.9)	
Multiplicity, *n* (%)				0.672
Solitary	40 (19.1)	32 (18.6)	8 (21.6)	
Multiple	169 (80.9)	140 (81.4)	29 (78.4)	
High-risk NMIBC, *n* (%)	164 (78.5)	129 (75)	35 (94.6)	0.009
BCG-refractory at 3 month, *n* (%)	28 (13.4)	-	28 (75.7)	<0.001
Preoperative NLR, mean ± SD	2.12 ± 0.99	1.97 ± 0.92	2.81 ± 1.02	<0.001
Preoperative NLR, mean ± SD	116.17 ± 49.61	111.91 ± 50.98	135.98 ± 37.27	0.007
Preoperative PLR, mean ± SD	5.42 ± 1.94	5.58 ± 1.99	1.70 ± 1.48	0.012
Interval time between BCG induction and TURBT, mean ± SD (months)	0.64 ± 0.45	0.65 ± 0.45	0.59 ± 0.43	0.546
Interval time between blood test and TURBT, mean ± SD (days)	14.04 ± 14.02	13.99 ± 13.35	14.27 ± 17.00	0.912

BCG = Bacillus Calmette–Guérin, BMI = body mass index, TURBT = transurethral resection of bladder tumor, CIS = carcinoma in situ, NLR = neutrophil-to-lymphocyte ratio, NMIBC = non-muscle-invasive bladder cancer, HG = high grade, BCG-unresponsive = T1HG recurrence at 3 months.

**Table 2 diagnostics-13-03114-t002:** Variables associated factors with non-complete response at 3 months after BCG induction.

	Univariable		Multivariable	
	OR (95% CI)	*p*	OR (95% CI)	*p*
Age > 70	1.101 (0.530–2.286)	0.797		
Sex (female vs. male)	1.352 (0.487–3.755)	0.563		
BMI (continuous)	1.033 (0.926–1.151)	0.564		
Urine cytology (neg vs. pos)	1.860 (0.797–4.340)	0.151		
Initial TURBT (no vs. yes)	0.615 (0.288–1.312)	0.208		
Size (≤3 cm vs. >3 cm)	1.591 (0.624–4.059)	0.331		
Multiplicity (Solitary vs. multiple)	1.207 (0.505–2.886)	0.672		
T stage (Ta vs. T1)	9.214 (2.689–31.572)	<0.001	9.100 (2.553–32.439)	0.004
Grade (low vs. high)	3.172 (1.173–8.576)	0.023		
Presence of CIS (no vs. yes)	1.310 (0.521–3.296)	0.566		
NLR (continuous)	2.147 (1.499–3.076)	<0.001	1.958 (1.331–2.882)	0.029
PLR (continuous)	1.009 (1.002–1.016)	0.009		
LMR (continuous)	0.748 (0.597–0.938)	0.012		

BCG = Bacillus Calmette–Guérin, BMI = body mass index, TURBT = transurethral resection of bladder tumor, CIS = carcinoma in situ, NLR = neutrophil-to-lymphocyte ratio.

## Data Availability

The dataset for the current study is available from the corresponding author on reasonable request.

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
