# Peer review of "Prognostic Role of Preoperative Neutrophil-To-Lymphocyte Ratio (NLR) and Recurrence at First Evaluation after Bacillus Calmette–Guérin (BCG) Induction in Non-Muscle-Invasive Bladder Cancer"

_diagnostics, 2023, doi:10.3390/diagnostics13193114_

Round 1
Reviewer 1 Report
I have reviewed the study titled ‘’Prognostic role of preoperative NLR and recurrence at first evaluation after BCG induction in non-muscle invasive bladder cancer’’. Many studies have been published on bladder tumors (invasive and non-invasive) regarding all of the hematological markers and NLR; the subject of the study is not interesting /original. Also, the data date of the study is not current. For these reasons, it does not make a different or meaningful contribution to the literature. I think the article should be rejected.
.
Author Response
Comments: I have reviewed the study titled ‘’Prognostic role of preoperative NLR and recurrence at first evaluation after BCG induction in non-muscle invasive bladder cancer’’. Many studies have been published on bladder tumors (invasive and non-invasive) regarding all of the hematological markers and NLR; the subject of the study is not interesting /original. Also, the data date of the study is not current. For these reasons, it does not make a different or meaningful contribution to the literature. I think the article should be rejected.
Response: We greatly appreciate the reviewer's time and thoughtful evaluation of our study. As your advice, we have prioritized updated our research by analyzing additional data (total n=273) for the two years until April 2020 within the tight revision timeline. The final analyzed total patient number was 209. We additionally analyzed recurrence and progression after BCG induction-only treatment according to primary TURB stage, as there are insufficient studies on the prognosis of BCG induction-only treatment.
There is also the significant issue of inconsistent definitions and terminology for BCG failure used by the EAU, AUA, and FDA. This discrepancy suggests the necessity for further research to establish standardized criteria for assessing BCG response. Lastly, there are many reports that NLR is associated with long-term prognosis of bladder cancer or BCG treatment, but our study is unique in its investigation of whether NLR is also associated with short-term response after BCG induction.
We kindly request the reviewer's reconsideration, taking into account the improvements we have made to address the mentioned concerns. Especially, we believe that our research makes a valuable contribution to the understanding of NLR's role in predicting short-term response after BCG induction. We hope that our revised study can be recognized for its significance in advancing knowledge in this field. We kindly request that if there are any further comments or suggestions.
Thank you for your consideration, and we look forward to seeing positive results.
Reviewer 2 Report
Dear authors, congratulations for your hard work. Bladdr cancer is one of the worst type of cancers, if i may say, considering its impact on patient's QoL. Ita has many variants and it is essential to have predictive tools at our disposal in order to identify high risk patients for recurrence, and to schedule the best approach.
Despite your effort, and your sincere approach to the subject, I have to stick to your comments about the limitations of your study.
2. it is a retrospective study with a small sample size and you propose that large scale well design studies must be conducted.
2. The definition and classification of BCG failure have not yet been fully established, each previous study on BCG failure also had differences in the study population. In some studies, CR and non-CR are used as one of the criteria to classify the BCG failure.
3. Third, the analysis did not include an assessment of long-term responses beyond 6 months after BCG induction. A large-scale, well-designed BCG treatment study and a clear definition of BCG failure is needed to improve the prognosis of bladder cancer.
Therefore, despite your good intentions, you haven't provided solid evidence to confirm your hypothesis, let alone you built your case upon a grey area, in specific what is the definition of BCG failure. To make a long story short, I propose you work on these issues and come back with a new improved version. Keep up your hard work!
Author Response
Response:
We greatly appreciate the reviewer's time and thoughtful evaluation of our study.
- As your advice, we have prioritized the update of our data within the tight revision timeline. Our research has been enhanced by analyzing two additional years of data (total n=273) through April 2020. The final analyzed total patient number was 209.
- There is also the significant issue of inconsistent definitions and terminology for BCG failure used by the EAU, AUA, and FDA. This discrepancy suggests the necessity for further research to establish standardized criteria for assessing BCG response. We described that T1 high-grade recurrence among non-complete response patients is associated with an unfavorable prognosis.
- We additionally analyzed recurrence and progression after BCG induction-only treatment according to primary TURB stage, as there are insufficient studies on the prognosis of BCG induction-only treatment. In situations where BCG treatment is limited, we have also demonstrated that initial Ta low-grade patients could cautiously consider BCG induction-only treatment.
We sincerely appreciate the reviewer's time and meticulous evaluation of our study. We kindly request that if there are any further comments or suggestions. Thank you for your consideration, and we look forward to seeing positive results.
Reviewer 3 Report
This study is based on patients with NMIBC who were retrospectively corrected at two centers in Korea. And this study examined whether the CR status at initial evaluation after BCG induction therapy (at 3 months) for the patients with intermediate or high risk NMIBC has an impact on disease prognosis. The study also showed that neutrophil-to-lymphocytes ratio (NLR) evaluated within 3 months prior to the first TUR affected disease prognosis, with an NLR of 2.42 or higher indicating a significantly worse prognosis.
This reviewer has several questions for the authors.
1 It states that NLR was measured within 3 months of the initial TUR-BT. If multiple measurements were taken, which values were used? Also, when was the mean time that 173 patients in the final analysis had their NLR measured? Since this is a retrospective study, this reviewer felt that the adoption of the values could be arbitrary on the part of the investigators. In order to demonstrate the fairness of the study, please provide details.
2 This study shows that NLR values affect the effectiveness of BCG induction. Please let me know if there are any reports in the literature that show whether NLR also affects BCG maintenance therapy. If not, I want to know the thinking/discussion from the authors.
3 This study shows several Kaplan-Meier curve diagrams. To me, the event and termination status is unclear due to poor resolution of the figure. Please add No. at risk to each figure.
4 I believe there are several spelling errors in the text (Ex. months -> monthss). In the results section, it is unclear whether the optimal NLR estimated from ROC is 2.4 or 2.42. Also, the abbreviation is used in the first mention of NLR (in the abstract). In the discussion, PLR and LMR are also abbreviations only. Please make the correction.
none
Author Response
We greatly appreciate the reviewer's time and thoughtful evaluation of our study.
- Response: We selected the NLR value taken on the day closest to the TURBT. The average time interval between NLR testing and TURBT was 14.04 ± 14.02 months. We have included this information in M&M and Table 1. Thank you for your suggestion.
-
Response: Thank you for comment. There are several studies on the prognostic role of NLR in patients receiving BCG maintenance therapy [1-4]. In these studies, NLR was reported to be a significant predictor of bladder cancer recurrence and progression in patients on BCG maintenance. BaÅŸer et al. reported significant differences in NLR values in patient classification according to EORTC progression and recurrence risk scores. And they reported a significant decrease in NLR values at preoperative, 3-month, and 12-month follow-up in 70 patients without recurrence and progression during BCG maintenance (2.31±1.03 vs. 2.24±1.17 vs. 2.13±1.10, p=0.002) [5].
These results are thought to be due to the association that activation of bladder cancer causes and maintain a systemic inflammatory response, and NLR is an index that reflects the systemic inflammatory response. We have further described this theory in the Discussion section as follows.
By understanding the immunologic characteristics of NLR and bladder cancer, it is possible to interpret how NLR functions as a predictor of prognosis in bladder cancer. In the tumor microenvironment, neutrophils play a role in promoting tumor progression by suppressing the anti-cancer immune system such as cytotoxic T lymphocytes and NK (natural killer) cells, as well as contributing to processes like angiogenesis and tumor cell proliferation. Conversely, lymphocytes infiltrate tumors and have an anti-tumor role. Therefore, NLR has been considered a biomarker that reflects systemic inflammatory responses (SIR) by utilizing these mechanisms to evaluate tumor activity. The growth of tumors is driven by a microenvironment that creates conditions favorable for tumorigenesis. This microenvironment involves various factors, including intrinsic inflammatory responses, angiogenesis, immune suppression, and cellular changes. Increased activity of cancer cells leads to SIR, and SIR in turn further promote cancer growth through activation of the pro-tumor responses, creating a vicious cycle. Bladder cancer has been reported to be highly immunogenic, evading normal immune responses. Considering the relationship between bladder cancer and immune response, NLR may be associated with bladder cancer long-term prognosis even with BCG maintenance therapy. NLR is known to have the advantages of being cost-effective, easy to perform, and less sensitive to physiological changes such as dehydration or physical activity compared to other indexes such as white blood cell count. Therefore, NLR is expected to be a valuable factor for predicting the prognosis of bladder cancer.
- Racioppi, M.; Di Gianfrancesco, L.; Ragonese, M.; Palermo, G.; Sacco, E.; Bassi, P.F. Can Neutrophil-to-Lymphocyte ratio predict the response to BCG in high-risk non muscle invasive bladder cancer? Int Braz J Urol 2019, 45, 315-324, doi:10.1590/S1677-5538.IBJU.2018.0249.
- Yuk, H.D.; Jeong, C.W.; Kwak, C.; Kim, H.H.; Ku, J.H. Elevated Neutrophil to Lymphocyte Ratio Predicts Poor Prognosis in Non-muscle Invasive Bladder Cancer Patients: Initial Intravesical Bacillus Calmette-Guerin Treatment After Transurethral Resection of Bladder Tumor Setting. Front Oncol 2018, 8, 642, doi:10.3389/fonc.2018.00642.
- Vartolomei, M.D.; Porav-Hodade, D.; Ferro, M.; Mathieu, R.; Abufaraj, M.; Foerster, B.; Kimura, S.; Shariat, S.F. Prognostic role of pretreatment neutrophil-to-lymphocyte ratio (NLR) in patients with non-muscle-invasive bladder cancer (NMIBC): A systematic review and meta-analysis. Urol Oncol 2018, 36, 389-399, doi:10.1016/j.urolonc.2018.05.014.
- Getzler, I.; Bahouth, Z.; Nativ, O.; Rubinstein, J.; Halachmi, S. Preoperative neutrophil to lymphocyte ratio improves recurrence prediction of non-muscle invasive bladder cancer. BMC Urol 2018, 18, 90, doi:10.1186/s12894-018-0404-x.
- BaÅŸer A. Does the Decrease in Neutrophil-lymphocyte Ratio after BCG Treatment Be a Prognostic Marker for NMIBC? J Urol Surg 2020, 7, 271-275.
If you have any more suggestions or questions, please let us know.
-
Response: Your input is greatly appreciated. If you have any more suggestions or questions, please don't hesitate to let us know.
-
Response: Thank you for your advice. We have made the corrections as suggested, including changing "monthss" to "months" and the cut-off values at 2.42. We have also fixed the issue where only the abbreviations PLR and LMR appeared. Your feedback is greatly appreciated.
We kindly request that if there are any further comments or suggestions. Thank you for your consideration, and we look forward to seeing positive results.